# Using an ideal observer analysis to investigate the visual perceptual efficiency of individuals with a history of non-suicidal self-injury when identifying emotional expressions

**Laura Ziebell** [1]*, **Charles Collin**[1], **Stéphane Rainville**[2], **Monica Mazalu**[1☯], **Madyson Weippert**[1☯¤]

**1** Department of Psychology, University of Ottawa, Ottawa, Canada, **2** VizirLabs Consulting, Chelsea, Québec, Canada

☯ These authors contributed equally to this work.
¤ Current address: Department of Nutritional Science, University of Toronto, Toronto, Canada
* lzieb079@uottawa.ca

**Data Availability Statement:** All relevant data are within the manuscript and its Supporting Information files.

## Abstract

Individuals who engage in non-suicidal self-injury (NSSI) often report significant interpersonal difficulties, with studies lending support to the idea of impaired social interactions. Perceptual processing deficits of facial expressions have also been associated with interpersonal difficulties, yet little research has assessed how individuals with a history of NSSI (HNSSI) process facial emotions. This study used an ideal observer analysis to assess emotion processing capabilities of these individuals. A total of 30 HNSSI and 31 controls were presented with static images of various facial expressions (fear, anger, disgust, happiness, sadness, surprise) at three intensity levels (50%, 75% and 100% emotion expressivity). Recognition of emotions were measured by signal-proportion thresholds, efficiency scores, and unbiased hit rate. Error responses were also recorded to investigate errors biases made by each group. No significant differences between HNSSI and controls were found in signal-proportion thresholds or efficiency scores. Decreased accuracy of HNSSI participants for recognizing fearful expressions was observed. An increased likelihood of mistaking angry for happy expressions and a decreased likelihood of mistaking sad for surprised expressions were recorded for the HNSSI group compared to controls. These findings provide support to the literature reporting deficits in accurate emotion identification for those engaged in NSSI behaviours.

## Introduction

Non-suicidal self-injury (NSSI) has been defined as the deliberate, self-inflicted injury to tissues of the body without suicidal intent [1, 2]. It has become a prevalent and clinically significant behaviour observed with increasing frequency over the past few decades among adolescents and young adults [3]. Furthermore, NSSI has been recognized as a significant public

**Funding:** This work was supported by funding from the Canadian Institutes of Health Research (http://www.cihr-irsc.gc.ca/e/193.html) to LZ under Grant 201310GAD; and the Natural Sciences and Engineering Research Council of Canada (https://www.nserc-crsng.gc.ca/index_eng.asp) to CC under Grant 2015-05067. The funders had no role in study design, data collection and analysis, decision to publish, or preparation of the manuscript. VizirLabs Consulting did not provide any financial contributions to this research.

**Competing interests:** Affiliation with VizirLabs Consulting does not alter our adherence to PLOS ONE policies on sharing data and materials. Stephane Rainville was paid to assist in programming the experiment and for creating scripts that aided in data aggregation. He did so based on instructions provided from the primary author.

health concern with estimates of lifetime prevalence rates between 5.5% in adults to 17.2% in adolescents [4], and 21–45% in clinical inpatient samples [5–7]. Since a clear association has been identified between NSSI and suicidal behavior [8–10], it is critical to understand factors that contribute to and predispose individuals to engage in this type of behaviour, with the hope of ultimately informing treatment and prevention strategies.

One of the most frequently reported functions for engaging in NSSI is related to emotion regulation [11–13]. Broadly defined, emotion regulation is the implicit and explicit effort to recognize, understand and manage one's own emotions and emotional expression [14–17]. An additional function reported for NSSI behaviours is related to efforts to attenuate, regulate or avoid negative emotional and cognitive states arising from challenging interpersonal interactions, as individuals who engage in NSSI often report significant interpersonal and social difficulties [18]. Those engaged in NSSI may do so as a means to communicate with others, particularly when less extreme attempts at communication fail to produce the desired results [19]. This is relevant to understanding the social functions that perpetuate NSSI because adolescents engaging in this behaviour have also been found to exhibit greater difficulty evaluating interpersonal situations [20] and display reduced vigilance to positive social information [21].

From an emotional intelligence framework, Yoo et al. [22] have examined the link between recognition and regulation of emotion, and concluded that emotion recognition may in fact be a precursor to emotion regulation. For instance, facial emotion perception is a central component to maintaining adequate social functioning, and if emotional facial expressions are not recognized accurately, then the ability to employ appropriate emotion regulation will ultimately be influenced. In this sense, accurate emotion recognition is critical, both within the self and in other people, to appropriately regulate emotional responses. Recent research has also linked engaging in NSSI with Alexithymia, an impairment in introspection about emotion that often co-occurs with poor emotion recognition in self and others [23] and poor non-verbal expressiveness [24]. In particular, individuals who have difficulty perceiving and understanding their own emotions may have difficulty navigating emotionally charged situations, especially when social interactions require the interpretation of non-verbal or subtle emotional cues. Hence, the ability to accurately infer facial emotions is essential for guiding one's own behaviours and regulating emotional states in various social situations. Supporting these concepts, recent research has also suggested that impaired ability to understand internal emotions may be linked to an impairment in emotion identification when faces are presented quickly or with visual degradation [23;25–29]

Results from our previous research [30] showed advantages in an emotion recognition task for a history of non-suicidal-self-injury (HNSSI) group, such as lower emotion intensity recognition thresholds and increased accuracy categorizing negative and ambiguous emotions, as compared to controls. Drawing from these results and the aforementioned studies, it is therefore of interest to further examine the emotional expression recognition capabilities of individuals with a HNSSI using a more psychophysical approach to this same question.

While Ziebell et al. [30] examined sensitivity in HNSSI in terms of emotional expression intensity, there are other dimensions along which sensitivity might vary. One of these dimensions is a simple perception signal-to-noise ratio. That is, subjects with high sensitivity to emotion expression might be expected to correctly categorize such expressions despite a high level of visual noise degrading the image. This type of sensitivity has been examined in a number of related contexts, including face detection in prosopagnosia [31] but to our knowledge, it has not been used to examine facial expression recognition. Understanding whether participants with HNSSI exhibit greater noise tolerance is useful in understanding this condition, as it will enable us to determine at what stage in visual processing differences in emotion recognition arise. While Ziebell et al. [30] showed that there was an advantage in negative emotion

recognition for HNSSI participants in terms of the minimum level of emotion intensity needed to identify an expression, this leaves open the question of whether this occurs due to an advantage in early visual processing, or by an advantage at some later higher-order stage of analysis.

This study therefore sought to investigate whether HNSSI participants exhibit an advantage in the signal-proportion threshold required for accurate emotion categorization, as compared to control participants. This was accomplished by presenting participants with images of emotionally expressive faces embedded in variable quantities of fractal luminance noise. Fractal luminance noise is defined as random variations in luminance wherein the contrast energy of the noise at each spatial frequency is equivalent to that in natural images. Put another way, the function of contrast energy by spatial frequency is defined by $E = 1/f^n$, where E is energy, f is frequency and n is an exponent, typically between 1 and 2.2. As such, the energy in the noise is proportional to that in the signal (i.e., the images of faces), with more energy at low spatial frequencies and less at higher ones. Measurement of the noise thresholds provides a means for assessing participants' sensitivity to emotional expression: The noisier the image with which they can correctly perform the emotion categorization task, the more sensitive they are to the information in the image of the emotion expression. The facial image and fractal noise are added together linearly to compose the final stimulus, and the amount of signal relative to the noise is expressed as a signal proportion. The addition of variable amounts of noise brings the amount of information available for a participant to use when identifying the emotion presented under exact experimental control. Importantly, the mixture of signal and noise was also varied such that overall image contrast always remained fixed: an increase in signal resulted in a decrease in noise and vice versa–a technique known as "titration" in the psychophysical literature. By design, the titration technique forces the signal and noise to covary in opposite directions, resulting in a ratio of signal energy to overall image energy–or "signal proportion"–bounded between 0.0 and 1.0. The more conventional metric in the literature of signal-to-noise ratio (SNR) is unbounded and typical of situations in which either signal or noise is fixed. Both metrics, signal-proportion and SNR, are related and can be mathematically re-expressed in terms of each other, so the choice of metric is simply dependent upon convenience. Titration is an elegant method that, in this particular case, also guards against any possibility that image contrast could somehow be a confounding variable associated with facial emotion.

As in our earlier experiment [30], our approach was designed to determine if NSSI participants show superior performance on this type of emotion recognition task compared to controls. Furthermore, because differences between NSSI and controls were observed at low expression intensities in our previous study, group differences in accuracy and errors will be investigated as a function of both emotion category and intensity.

In addition to measuring noise thresholds, the performance of participants will be subjected to an ideal observer analysis. This kind of analysis provides an informative approach to understanding the visual recognition of facial emotions by humans. The central concept of an ideal observer analysis is the "ideal observer algorithm" (IO), which is a computer program that performs a given visual task, like emotion recognition, in a mathematically optimal fashion given the available information, and within specified constraints [32]. This is not to say that the ideal observer will perform without error, but rather that it will perform at the physical limit of what is possible given the available information in the facial emotion stimulus [32]. When an ideal observer makes a mistake, it is generally due to the complexity and uncertainty in the visual environment, such as inherent noise in light or the signal used by the ideal observer [32]. The ideal observer algorithm first determines its own performance on the emotion recognition task, which is then applied as a baseline against which to compare the performance of human participants. The ratio of human performance to ideal performance, known as "efficiency,"

factors out the effects of variation in information content, and yields a pure measure of a human observer's relative ability to make use of the information available [33]. This provides a means by which to determine which emotions are objectively easier to detect and which are more difficult, clarifying interpretation of human performance data. Note that ideal observers are only ideal within the narrow scope for which they are designed. For instance, the ideal observer in this experiment operates purely on the pixel-by-pixel correspondence between the stimulus and the set of stored noiseless templates; however, absolute ideal performance is not the goal. The goal here is that an ideal observer is simply a model that captures important information about a stimulus set and allows one to evaluate human performance in the context of task difficulty, or in this particular case, by taking into account the possibility that some emotions may be intrinsically more difficult to detect than others regardless of who is doing the detecting.

Expanding on the results of Ziebell et al., [30], it is expected that individuals with a HNSSI will be more sensitive to negative emotions. In particular, it is anticipated that they will have lower signal-proportion detection thresholds and higher efficiency scores for the emotions of fear, anger, disgust and sadness at all emotion intensities compared to controls. It is also expected that the discrepancy between signal-proportion detection thresholds for the HNSSI and controls will be most pronounced at the lowest (50%) emotion intensity, as emotion identification is more challenging at this intensity level, and the HNSSI group's greater identification sensitivity may become more evident. More generally, comparing human performance to the ideal observer performance for detecting various facial expressions will help elucidate if differences in the ease of detection are the result of differences inherent to the stimuli, or if they are attributable to variations in human higher-order processing of emotions. For example, if detecting happy faces is fundamentally easier due to information in the stimuli and unrelated to characteristics of the observer, calculating an efficiency score will eliminate enhanced detection of happiness and theoretically equate its detection threshold with all other emotion-detection thresholds.

It is also expected that the HNSSI participants will display greater accuracy in emotion identification for negative emotions, as assessed by the *unbiased hit rate* [34]. Calculating a simple hit rate is valid for measuring overall accuracy when considering all emotions combined. However, when measuring the accuracy of specific emotions, the hit rate does not consider "false alarms", or biases in the use of one or more response categories [34]. In other words, in order to measure recognition accuracy by a participant for a given emotion, the number of misses (i.e., mislabelling disgust as anger), as well as the number of false alarms (i.e., incorrectly labeling another emotion as anger) should be taken into account. Failure to do so could lead to less precise calculations of accuracy rates. Thus, in the present study the unbiased hit rate was primarily used as a measure of accuracy.

Finally, error patterns will be compared between the HNSSI and control groups to determine if biases in error patterns are evident. Any differences may further highlight emotion categorization differences between the HNSSI and control groups.

## Materials and methods

### Participants

The study sample is composed of young adults (between 17 to 24 years of age) recruited from an undergraduate subject pool at the University of Ottawa. The University of Ottawa Ethics Review Board approved this research study. Approval was also granted by the Research Ethics Committee to include minors (participants under age 18) in the study without parental or guardian consent. A total of 61 participants (30 HNSSI and 31 control) were recruited. Refer

**Table 1. Participant demographics.**

| Variable | HNSSI group(*n* = 30) | Control group(*n* = 31) | *p* (d.f.) |
|---|---|---|---|
| Age: years | 19.00 ± 1.84 | 19.23± 1.56 | 0.61 (59) |
| Sex: male | 10% (3) | 19% (6) | 0.31 (59) |
| Ethnicity: White | 73% (22) | 58% (18) | 0.99 (59) |
| Past Diagnosis: | | | |
| Depression | 37% (11) | 6% (2) | <0.01 (59) |
| GAD | 27% (8) | 7% (2) | 0.03 (59) |
| PTSD | 3% (1) | 3% (1) | 0.99 (59) |
| OCD | 7% (2) | 0% (0) | - |
| Other | 13% (5) | 3% (1) | 0.04 (59) |
| None | 43% (13) | 84% (26) | <0.01 (59) |

to Table 1 for demographic characteristics of the sample by group. No differences were found between HNSSI and the control groups for age, sex or ethnicity. Not surprisingly, the HNSSI group had higher rates of comorbid depression, anxiety and other past diagnoses compared to the control group.

## Eligibility criteria

Pre-screening questions, administered through the recruitment website, were used to identify a subset of the population who had a HNSSI, but reported no history of a Borderline Personality Disorder (BPD) diagnosis. For inclusion in the HNSSI group, a participant had to have engaged in intentional self-inflicted injury to the surface of his or her body at least 5 times in their lifetime, with the expectation that the injury would lead to only minor or moderate physical harm (i.e., no suicidal intent). The pre-screening question read, "Have you ever intentionally self-inflicted damage to the surface of your body to cause bleeding, bruising, or pain (e.g., cutting, burning, stabbing, and/or hitting), without the intent to kill yourself? Please note that this does not include ear piercing, tattooing, circumcision, or cultural healing rituals." Potential responses included Never; Once; 2–4 times; 5 or more times. Only individuals who responded "Never" or "5 or more times" were screened in to participate as controls or HNSSI respectively. Exclusion criteria for both the NSSI and control groups included a self-reported diagnosis of Borderline Personality Disorder. The pre-screening question read, "Have you ever been diagnosed with Borderline Personality Disorder? Yes/No." Additionally, individuals who reported to have engaged in NSSI "5 or more times" on the pre-screening question, but failed to report NSSI behavior in either of the administered NSSI questionnaires (see measures below) were also excluded from the study. Furthermore, administered questionnaires were only validated for research purposes in English, therefore, participants were excluded if they are unable to read and understand English.

All HNSSI participants reported having engaged in intentional self-inflicted injury to the surface their body at least 5 times or more within their lifetime. The majority of HNSSI reported thinking about self-injuring within the past month 60% (*n* = 18), and 17% (*n* = 5) had actually engaged in the behavior within the month. Additionally, 70% (*n* = 21) of the HNSSI participants reported thinking about self-injuring within the past 6 months and half the sample reported actually engaging in self-injuring within that timeframe 50% (*n* = 15).

## Measures

**Socio-demographic questionnaire.** This demographic questionnaire collected standard participant information such as age, gender, primary language, ethnicity, education, and

current or past mental health diagnosis. This information was used to descriptive compute statistics of the NSSI and control groups.

**The Ottawa Self Injury Inventory.** This questionnaire (OSI—Functions 1.1) assessed self-injurious behaviours and their functions. This scale provided cumulative scores for the subscales of internal emotional regulation, external emotional regulation, social influence, and sensation seeking [35].

**The Inventory of Statements About Self-Injury.** This questionnaire (ISAS—Section II) assessed an individual's reasons for engaging in self-injurious behaviours. This scale provided a cumulative score for subscales of interpersonal functions of NSSI (i.e., autonomy, interpersonal boundaries, interpersonal influence, peer-bonding, self-care, revenge, sensation seeking, and toughness) and Intrapersonal Functions (i.e., affect-regulation, anti-dissociation, anti-suicide, marking distress, self-punishment) of NSSI [36].

## Procedure

**Consent.** Prior to the task, and after having the study described to them verbally, participants read and signed an informed consent form.

**Questionnaires.** All participants completed the socio-demographic questionnaire. Two additional questionnaires, the Ottawa Self Injury Inventory (OSI) and the Inventory of Statements About Self-Injury (ISAS), were completed by the HNSSI group only. If any HNSSI participants reported suicidal ideation or severe harm, by endorsing Q3, Q4, Q5, or Q6 on the OSI, a suicide protocol was implemented. Additionally, the NSSI participants were all given a list of resources in the event they wished to seek further psychological support. Questionnaires were administered prior to starting the experiment to identify any participants actively engaged in suicidal ideation, and to administer the appropriate suicidal assessment protocol to these vulnerable participants. Moreover, approximately 20 minutes elapsed between completion of the questionnaires and the beginning of study administration to mitigate any emotional arousal arising from completing the questionnaires that might otherwise have affected participant performance on the study.

**Stimuli.** Participants were presented with static grey scale images of faces expressing one of six emotions (happiness, sadness, anger, fear, disgust or surprise) at varying degrees of emotion intensity (50%, 75% or 100%). Stimuli were masked with varying degrees of fractal luminance noise. The energy vs. spatial frequency function of the fractal noise matched that of the face image. The amount of fractal noise, added to the otherwise noiseless stimuli, effectively controlled the difficulty of the task (i.e., "signal-to-noise ratio"), such that more noise destroyed available information or, equivalently, corrupted the signal and increased the difficulty of emotion detection. As previously mentioned, the contrast of the image (root mean square) was kept constant regardless of how the proportion of signal and noise energy was titrated.

Although signal proportion (i.e., the relative amount of signal and noise contributing to the stimulus) varied from trial to trial, the signal and noise components always had identical so-called pink–or "fractal"–spectral profiles where contrast energy is concentrated at lower spatial frequencies on a per-frequency basis, but distributed equally across constant-octave frequency bands. A total of 58 identities (29 male and 29 female) from the Karolinska Directed Emotional Faces [37] database were used for this task; however, each participant was only tested with a single identity. This was done to avoid interactions with facial identification. An oval window was created around each face to best approximate its contours. The window was then smoothed and window edges were slightly blurred by applying a low-pass filter to produce a gradual transition between the face image and the gray background (Fig 1). There were 58 identities x 6 emotions x 3 emotional intensities for a total of 1044 face images.

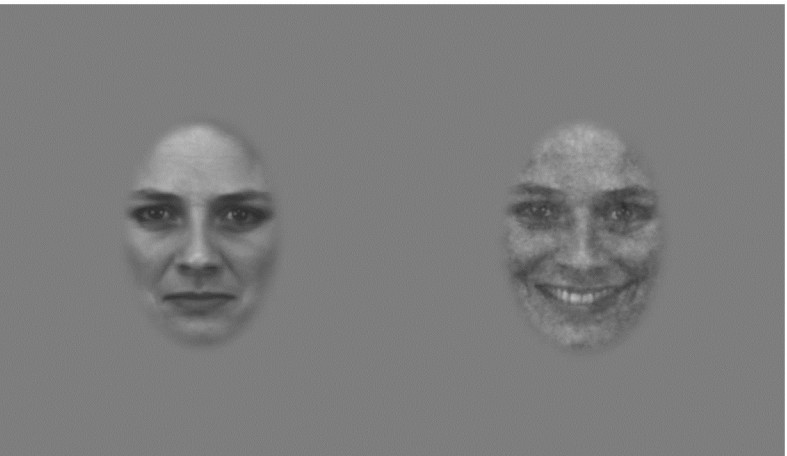

**Fig 1. Two examples of stimulus used for emotion categorization task.** The first image is an example of a disgusted face at 50% intensity without any fractal noise. The second image is an example of a happy face at 100% intensity embedded in fractal noise intended to partially mask the image. These images were modified from the Karolinska Directed Emotional Faces database, with "AF01DIS" and "AF01HAS" displayed. Reprinted from Lundqvist, D., Flykt, A., & Öhman, A. under a CC BY license, with permission from the Karolinska Institutet original copyright (1998).

To introduce noise into the image, the face and the fractal noise were added linearly to create the final stimulus. The amount of signal (face image) relative to the noise is expressed as a signal proportion, or Var(Signal)/(Var(Signal) + Var(Noise)) that varied between 0 (noise only) to 1 (signal only). Note that the alternative metric for signal-to-noise ratio (SNR) previously referred to is simply Var(Signal)/Var(Noise), but signal proportion was favoured largely as a matter of convenience. The signal proportion was expressed in *Log2* units because human performance tends to be linear with respect to the logarithm of signal proportion. This means that a signal-only stimulus (no noise) has a signal-proportion of *Log2* (1) = 0, an equal mixture of signal and noise has a signal-proportion of *Log2* (0.5) = -1, and a noise-only stimulus would have a proportion of *Log2* (0) = -∞. However, a noise only stimulus is uninformative for this task, as there is no face information for the participant to detect, and was not used as a stimulus.

**Human performance of emotion recognition task.** Participants began the computer task with practice trials in which they were asked to identify each of the six basic emotions described by Ekman [38] (happiness, sadness, anger, fear, disgust and surprise) through a computer-based 6-alternative forced-choice (6AFC) procedure. A total of six practice trials were presented, one for each of the six emotions presented during the experiment. These practice images were presented at full emotion intensity and without fractal noise to facilitate learning of the corresponding keyboard responses. These practice trials not only helped the participant associate the emotions with the correct keyboard responses, but also helped to familiarize them with the various emotion expressions of their selected stimulus identity.

Once the practice trials were completed, participants began the experimental trials. On each trial, a participant was presented with a face image. The identity of the model presented in the image did not change for a given subject. That is, each subject was tested on a single individual model. The image varied in emotion intensity (50%, 75%, or 100%), emotion category (sad, happy, angry, afraid, surprised or disgusted), and degree of fractal noise. The design matrix of the experiment included 6 emotions X 3 intensities X 50 repetitions for a total of 900 trials. Eighteen randomly-interleaved staircases, one for each emotion x intensity, ran in parallel such that each staircase updated the stimulus' signal-proportion on each trial to converge

toward a threshold in a one-up-one-down fashion. For example, if a participant correctly categorized the stimulus "75% sadness" at a given signal-proportion, then the degree of fractal noise obscuring the image was increased in the next presentation of that stimulus; conversely, if they misidentified the emotion, the degree of fractal noise obscuring the image decreased during the next presentation of that stimulus (ie, a one-up one-down procedure). By definition, the one-up one-down procedure converges on the 50% correct point and clusters observations sufficiently near the more theoretically informative performance level in a 6AFC task, 58.3% which is halfway between chance (1/6 = 16.7%) and ceiling (100%). Once data were gathered, a separate psychometric function was fitted to each unique combination of emotion and intensity, whereby signal-proportion thresholds were computed corresponding to 58.3% performance.

**Ideal observer performance of emotion recognition task.**    In addition to gathering data from human observers, an Ideal Observer Algorithm also performed the task. This was a computer program that performed the same task as human observers, but did so by making ideal use of the visual information available in the images. Specifically, the Ideal Observer program worked as follows: For each trial, the algorithm searched through the database of all the images used in the experiment to find the best match with the image presented in the trial. The best match was determined by calculating the correlation between pixel values in the presented stimulus image and pixel values in each image in the database. In noiseless conditions, where the pixel values in the image presented for a trial are identical to the pixel values in the image from the database yield a perfect correlation ($r = 1$, i.e. the two images are identical) and lower r values for all other database images, thus performing the task perfectly with 100% accuracy.

The next step was to add noise to the images. In a similar fashion to what was done with human observers, variable amounts of fractal noise were added to the images, until the ideal observer's performance fell to the 58.3% accuracy threshold. The amount of stimulus signal needed to do this was the variable of interest, and was labeled the signal proportion threshold. In theory, this threshold represents the minimum amount of signal (or, equivalently, the maximum amount of noise) that any perceptual system could tolerate in doing the task. However, it should be noted that an Ideal Observer is only an ideal *observer*, not an ideal thinker or performer. That is, the IO makes perfect use of all the information in the stimulus, but nothing else. It has no knowledge, for instance, of shape-from-shading, or average facial structure, or 3D shape. Thus, it was possible for a human to outperform the ideal observer if the human's higher-order knowledge allowed them to use more than simply perceptual information.

## Data analysis

**Power analysis.**    In order to ensure sufficient power, a priori power analysis was used to determine the sample size required per group. In a study that examined sensitivity to facial expressions of emotion in BPD using static pictures of facial affect [39], a large Cohen's d effect size of 0.8 (Effect size f = 0.4) was reported for greater sensitivity across the six emotions examined (anger, disgusts, fear, happiness, sadness, surprise) [40]. However, there is little to no available literature to guide an estimate of effect size for thresholds or efficiency scores in a comparable population. Our efficiency scores are a ratio calculation, and thus prone to greater error, so a more conservative estimate of effect size was selected as a prudent measure. Assuming a small effect size f = 0.10 (Cohen's d of 0.2), power of 0.8 with α of 0.05 to compute sample size using GPower 3.1.9.2 for an ANOVA Repeated-Measures, within-between interaction of a 2 (NSSI and Control) X 6 (happy, sad, anger, fear, disgust, and surprise) X 3 (50%, 75%, 100%) design, the estimated total sample size is 56 participants. Hence, the total collected sample size

of 61 (30 NSSI and 31 Control) is likely adequate for the interpretation of any non-significant findings.

## Results

### Threshold and efficiency

Threshold and efficiency scores were analyzed using planned contrasts [41] to test the above stated hypotheses. In order to obtain the error terms for the planned contrasts [41], the data were subjected to mixed factorial ANOVAs, with group (NSSI and control) as the between-subjects factor, and stimulus emotion (sad, disgusted, surprised, fearful, angry and happy faces) and intensity (50% intensity, 75% intensity and 100% intensity) as the within-subjects factors (Table A and Table B in S1 File).

Due to the nature of participants' responses and the minimum number of trials presented per emotion, calculating a threshold and efficiency score for all participants was not possible, resulting in an 8.4% loss of data for both variables. The chi-squared statistic referred to as Little's MCAR test was used to test whether values were missing completely at random [42]. The MCAR test used for the missing threshold values resulted in $\chi^2$ (462, $N$ = 61) = 507.72, $p$ = .070, and $\chi^2$ (462, $N$ = 61) = 498.50, $p$ = .117 for the missing efficiency values, indicating that data were indeed missing at random (i.e., no identifiable pattern existed in the missing values). This finding means that our data meet the assumptions for the application of imputation methods to fill in missing data.

To maximize available data, the method of data imputation known as Multiple Imputation (MI) was selected as an appropriate technique for addressing the missing values in this circumstance. MI is superior to a single imputation because it makes repeated draws from a model of the distribution of variables that have missing values to create several complete datasets. These datasets can then be analyzed in parallel. Variations in outcome between the datasets reflect uncertainty from the imputation process itself [43]. As suggested by White, Royston, and Wood [44], the number of imputations should be greater than or equal to the percentage of missing observations to ensure an adequate level of reproducibility. Since the dataset had 8.4% missing data for both variables, 9 imputations were selected to produce a total of 9 imputed datasets for each variable in accordance with this suggestion. The MI procedure was carried out with IBM SPSS Statistics 24. Prior to imputation, the dataset was assessed for univariate outliers. Due to the ambiguity of the sample size created by using the multiple imputation procedure, a standard z-score cut-off of above 2.5 *SD*, was used to identify outliers, which were winsorized by having their scores changed to the closest non-outlying score [45,46]. Overall, 3.01% of threshold scores and 2.28% of efficiency scores were considered univariate outliers and winsorized.

Typically, multiple imputation combines output across all imputed datasets, and the results are pooled for interpretation, which also produces confidence intervals. However, because no explicit rules have been defined for pooling *F*-tests of (repeated-measures) analysis of variance [47] and this function is not available for an ANOVA calculation in IBM SPSS Statistics 24, the data for the median error term of the 9 imputed datasets was used to obtain the mean squared value for the contrast calculations. These calculations were also repeated with the lowest and highest mean squared value produced by the multiply imputed datasets to create an upper and lower bound for the analysis, and to act as a confidence interval.

Considering a relatively large number of planned contrasts (18), and to account for the increased probability of making a type I error, the alpha level was adjusted by using a *Bonferroni approach* as suggested by Rosenthal and Rosnow [41] at $\alpha$ = 0.05/18 = 0.003. Given that group sample sizes differed slightly, the harmonic mean between sample sizes across both groups was used in all contrast analyses [41]. Effect sizes for the contrasts were also measured

and reported as *r* [41]. Values of *r* ranging from 0.10, 0.30 and 0.5 are considered small, medium and large effect sizes respectively [41]. The results of the contrast analyses are reported here and the ANOVA tables are presented in the Appendix.

## Emotion detection scores for threshold and efficiency

Results from the planned contrasts showed no consistent evidence of group differences for threshold scores between the HNSSI and control group (see Table 2). This result remained consistent when testing contrasts at the upper and lower bound of the mean squared value obtained for the 9 imputed datasets. That is, regardless of whether a strict or liberal criterion was applied to our contrast analyses, the two groups displayed comparable threshold scores for each emotion and across all intensities presented. We note that this same result is obtained whether one applies the Bonferroni correction or not, showing that it is not the result of overly conservative statistical procedures.

Results from the planned contrasts for efficiency also showed no consistent evidence of group differences between the HNSSI and control group (see Table 3). While there were differences at the 0.05 level for 75% happy and 50% sad stimuli, the effect sizes were small and these effects were not consistent across intensities, nor across negative emotions. Moreover, the Bonferroni correction resulting in an adjusted $\alpha$ level of $p < 0.003$ made it such that these results are in fact non-significant. Again, the same pattern of results remained consistent when testing contrasts at the upper bound of the mean squared value for the 9 imputed datasets, and the lower bound also did not reach a significance level of $p < 0.003$.

## Emotion recognition accuracy at various intensities

Participants' accuracies for recognizing emotions at 50%, 75% and 100% intensities are reported in Table 4. The unbiased hit rate [34] was calculated for each of the six emotion

**Table 2. Results of planned comparisons for threshold.**

| Emotion | Intensity | NSSI(Mean ± SE) | Control(Mean ± SE) | Effect Size (r) | p |
|---|---|---|---|---|---|
| Fear | 50% | 0.443 ± 0.045 | 0.382 ± 0.034 | 0.098 | 0.111 |
| | 75% | 0.321 ± 0.039 | 0.287 ± 0.022 | 0.062 | 0.313 |
| | 100% | 0.286 ± 0.037 | 0.239 ± 0.020 | 0.088 | 0.156 |
| Anger | 50% | 0.241 ± 0.019 | 0.243 ± 0.019 | 0.003 | 0.961 |
| | 75% | 0.188 ± 0.015 | 0.203 ± 0.017 | 0.028 | 0.645 |
| | 100% | 0.167 ± 0.016 | 0.193 ± 0.027 | 0.048 | 0.441 |
| Disgust | 50% | 0.360 ± 0.037 | 0.406 ± 0.035 | 0.075 | 0.227 |
| | 75% | 0.302 ± 0.037 | 0.352 ± 0.040 | 0.093 | 0.133 |
| | 100% | 0.260 ± 0.035 | 0.310 ± 0.039 | 0.091 | 0.138 |
| Happy | 50% | 0.148 ± 0.011 | 0.173 ± 0.011 | 0.040 | 0.513 |
| | 75% | 0.099 ± 0.004 | 0.106 ± 0.007 | 0.012 | 0.849 |
| | 100% | 0.075 ± 0.006 | 0.085 ± 0.006 | 0.019 | 0.763 |
| Sad | 50% | 0.225 ± 0.019 | 0.201 ± 0.016 | 0.038 | 0.536 |
| | 75% | 0.174 ± 0.012 | 0.172 ± 0.012 | 0.003 | 0.960 |
| | 100% | 0.166 ± 0.013 | 0.165 ± 0.010 | 0.002 | 0.965 |
| Surprise | 50% | 0.364 ± 0.042 | 0.325 ± 0.038 | 0.063 | 0.308 |
| | 75% | 0.236 ± 0.023 | 0.176 ± 0.015 | 0.111 | 0.071 |
| | 100% | 0.168 ± 0.016 | 0.136 ± 0.013 | 0.059 | 0.338 |

*Note*. Mean and SE are calculated based on the average of the 9 imputed datasets. Effect size and p-values are calculated using the median mean square error and df of the 9 imputed datasets.

**Table 3. Results of planned comparisons for efficiency.**

| Emotion | Intensity | NSSI(Mean ± SE) | Control(Mean ± SE) | Effect Size (r) | p |
|---|---|---|---|---|---|
| Fear | 50% | 0.473 ± 0.072 | 0.586 ± 0.066 | 0.064 | 0.190 |
| | 75% | 0.464 ± 0.056 | 0.436 ± 0.065 | 0.016 | 0.742 |
| | 100% | 0.389 ± 0.052 | 0.357 ± 0.045 | 0.007 | 0.924 |
| Anger | 50% | 0.753 ± 0.112 | 0.687 ± 0.097 | 0.038 | 0.445 |
| | 75% | 0.578 ± 0.062 | 0.617 ± 0.075 | 0.022 | 0.646 |
| | 100% | 0.635 ± 0.135 | 0.471 ± 0.053 | 0.040 | 0.625 |
| Disgust | 50% | 0.603 ± 0.080 | 0.477 ± 0.041 | 0.071 | 0.149 |
| | 75% | 0.451 ± 0.042 | 0.443 ± 0.038 | 0.004 | 0.928 |
| | 100% | 0.397 ± 0.034 | 0.344 ± 0.042 | 0.013 | 0.873 |
| Happy | 50% | 1.048 ± 0.095 | 0.995 ± 0.099 | 0.030 | 0.541 |
| | 75% | 0.830 ± 0.057 | 1.007 ± 0.079 | 0.099 | 0.042 |
| | 100% | 0.914 ± 0.081 | 0.970 ± 0.096 | 0.013 | 0.868 |
| Sad | 50% | 0.971 ± 0.092 | 1.213 ± 0.124 | 0.136 | 0.005 |
| | 75% | 0.731 ± 0.067 | 0.719 ± 0.063 | 0.007 | 0.888 |
| | 100% | 0.533 ± 0.049 | 0.525 ± 0.046 | 0.002 | 0.981 |
| Surprise | 50% | 0.490 ± 0.083 | 0.494 ± 0.058 | 0.002 | 0.961 |
| | 75% | 0.465 ± 0.070 | 0.469 ± 0.057 | 0.002 | 0.967 |
| | 100% | 0.422 ± 0.055 | 0.544 ± 0.095 | 0.030 | 0.718 |

*Note.* Mean and SE are calculated based on the average of the 9 imputed datasets. Effect size and p-values are calculated using the median mean square error and df of the 9 imputed datasets.

**Table 4. Unbiased hit rate for recognition of facial expressions presented at 50%, 75% and 100% intensity.**

| Emotion | Intensity | NSSI(Mean ± SE) | Control(Mean ± SE) | Effect Size (r) | p |
|---|---|---|---|---|---|
| Fear | 50% | 0.151 ± 0.023 | 0.189 ± 0.022 | 0.115 | 0.037 |
| | 75% | 0.194 ± 0.024 | 0.265 ± 0.023 | 0.211 | $< 0.001^*$ |
| | 100% | 0.212 ± 0.023 | 0.273 ± 0.023 | 0.182 | $< 0.001^*$ |
| Anger | 50% | 0.272 ± 0.012 | 0.253 ± 0.014 | 0.058 | 0.295 |
| | 75% | 0.281 ± 0.011 | 0.264 ± 0.018 | 0.052 | 0.349 |
| | 100% | 0.286 ± 0.010 | 0.274 ± 0.017 | 0.036 | 0.506 |
| Disgust | 50% | 0.250 ± 0.025 | 0.263 ± 0.020 | 0.039 | 0.473 |
| | 75% | 0.251 ± 0.023 | 0.244 ± 0.023 | 0.021 | 0.699 |
| | 100% | 0.257 ± 0.020 | 0.252 ± 0.025 | 0.015 | 0.783 |
| Happy | 50% | 0.361 ± 0.012 | 0.367 ± 0.015 | 0.018 | 0.741 |
| | 75% | 0.356 ± 0.013 | 0.356 ± 0.013 | 0.000 | 1.000 |
| | 100% | 0.362 ± 0.012 | 0.363 ± 0.013 | 0.003 | 0.956 |
| Sad | 50% | 0.215 ± 0.012 | 0.244 ± 0.014 | 0.088 | 0.110 |
| | 75% | 0.255 ± 0.014 | 0.288 ± 0.013 | 0.100 | 0.069 |
| | 100% | 0.290 ± 0.015 | 0.297 ± 0.015 | 0.021 | 0.699 |
| Surprise | 50% | 0.313 ± 0.024 | 0.293 ± 0.019 | 0.024 | 0.659 |
| | 75% | 0.340 ± 0.021 | 0.332 ± 0.019 | 0.019 | 0.776 |
| | 100% | 0.346 ± 0.020 | 0.335 ± 0.017 | 0.033 | 0.544 |

$^*$ Significant at the $p < 0.003$ level

expressions to measure recognition accuracy. This index was selected because it is a more appropriate measure than hit rate, due to the fact that it accounts for participants' non-uniform use of other emotion categories and thus mitigates response bias. The datasets were screened for assumptions and assessed for univariate outliers. Since no missing data were present in the accuracy scores, and a full dataset was analyzed, an outlier was defined as having a z-score above 2.43 *SD* in accordance with the Van Selst and Jolicoeur [48] recommendations for the above stated sample size, and outliers were winsorized by having their scores changed to the closest non-outlying score [45–46]. Overall, 1.3% of the NSSI data and 3.2% of the control data were considered univariate outliers and winsorized. Normality was assessed using Shapiro-Wilk's test ($p < .05$), and skewness and kurtosis scores fell within a ±2 criterion range [49–50]. In the NSSI group, non-normality was detected for fear and disgust at 50%, 75% and 100% intensity. In the control group, non-normality was detected for fear at 50%, 75% and 100% and anger and disgust at 50%. Applying an Arcsine transformation did not improve the distribution and we therefore opted to continue with the analysis using the non-transformed data.

In order to obtain the error terms for the planned contrasts [41] the data were subjected to mixed factorial ANOVAs, with group (history of non-suicidal self-injury group and control group) as the between-subjects factor, and stimuli type (sad, disgusted, surprised, fearful, angry and happy faces) and intensity (50%, 75%, and 100%) as the within subject factors (Table G in S1 File).

As with our previous analyses, a relatively large number of planned contrasts (18), were performed, so to decrease the probability of making a type I error, the alpha level was again adjusted with the *Bonferroni approach* as suggested by Rosenthal & Rosnow [41] at $\alpha = 0.05/18 = 0.003$. Effect sizes for the contrasts were also measured and reported as *r* [41]. Results from these planned contrasts for unbiased hit rate show that the HNSSI group were less accurate at identifying fear at the 75% and 100% intensity level. This trend was also observed in the fear accuracy scores at 50% intensity. Accuracy scores obtained through hit rate were also calculated and compared between groups, which showed the same pattern of results consistent with those obtained from the unbiased (Tables D and E in S1 File). There were no other consistent or significant results observed for other emotions or across other intensity levels.

## Group differences in error patterns

The next analysis was intended to further examine the nature of group differences by assessing the types of errors made by each group when misidentifying an emotion. Accordingly, a confusion matrix analysis with the number of errors was performed to determine if participants were mistaking one emotion for another as each emotion was presented at the three intensities.

Data were first assessed by the Shapiro-Wilk test, which is most appropriate for smaller sample sizes, to assess normality (Table F in S1 File). Although many of the error responses appeared non-normally distributed according to this conservative statistic, the majority of data fell within a ±2 criterion range for skewness and kurtosis (Table G in S1 File) and were thus considered normally distributed [49–50]. Visual inspection of the distributions revealed that responses were similarly slightly skewed in a positive direction. Additionally, data were assessed for outliers as defined by having a z-score above 2.43 *SD* in accordance with the sample size [48]. For each of the emotions presented, 3.1% of the NSSI data and 4.3% of the control data were considered univariate outliers for fear; 2.9% of the NSSI data and 3.2% of the control data for anger; 4% of the NSSI data and 3.7% of the control data for disgust; 2.2% of the NSSI data and 2.6% of the control data for happy; 2.4% of the NSSI data and 2.6% of the control data

for sad; and 3.1% of the NSSI data and 1.5% of the control data for surprised were considered univariate outliers. These outliers were winsorized by having their scores changed to the closest non-outlying score [45–46].

## Error analysis at 50%, 75% and 100% emotion intensity

Error pattern analysis for each of the six emotions across intensities showed occasional significant results, however, few consistent patterns were observed. Of note, the HNSSI group was significantly less likely to confuse sad expressions with the emotion of surprise at 50% and 75% intensity compared to the control group, a statistically significant difference of $t(59) =$ -2.24, $p = .019$, $d = 0.62$, and $t(59) = -2.67$, $p = .009$, $d = 0.69$ respectively. When surprise was presented at 100% intensity, the reverse was true, as the HNSSI group was significantly less likely to mistake surprise with the emotion of sadness compared to controls, a statistically significant difference of $t(59) = -2.43$, $p = .018$, $d = 0.62$.

Additionally, a pattern was noted between the emotions of anger and happiness, with the HNSSI group being significantly more likely to mistake angry expressions at 50% intensity for expressions of happiness compared to controls, a significant difference of $t(49.82) = 2.20$, $p = .033$, $d = 0.57$. The HNSSI group was also more likely to mistake happy expressions at 100% intensity for the emotion of anger compared to controls, a significant difference of $t(59) = 2.19$, $p = .033$, $d = 0.56$. When collapsed across emotion intensity, some of these patterns remained. For example, the HNSSI group was significantly less likely to confuse sad expressions with the emotion of surprise and significantly more likely to confuse angry expressions with the emotion of happiness, a statistically significant difference of $t(59) = -2.31$, $p = .024$, $d = 0.56$ and $t(59) = 2.18$, $p = .033$, $d = 0.59$ respectively. Refer to the Supplemental Information for a full error analysis summary (S1–S9 Figs).

## Discussion

The present study investigated if HNSSI participants showed advantages in the signal-proportion threshold required for accurate emotion categorization compared to control participants. Furthermore, participants' performance was subjected to an Ideal Observer analysis to determine their efficiency ratio, which factors out variation in information content to yield a pure measure of an observer's performance. This model used a simple algorithm to calculate the dot product of the stimulus image with each potential available image to identify the maximally likely match, thus optimizing the available image information. Results from this study revealed no differences between HNSSI and control participants for signal-proportion detection threshold or efficiency ratios for any of the emotions presented at all three intensities. These results were obtained despite adequate samples sizes determined by a priori power analyses, sufficient power, and focused statistical analyses. No observed differences between groups may imply that, for a lower level of visual analysis, a history of engaging in NSSI behaviour has no effect on emotion detection thresholds. Likewise, having engaged in NSSI behaviours does not appear to relate to a greater efficiency in detecting emotional faces, as calculated through ideal observer analysis. These results are in contrast to the results obtained by Ziebell et al., [30], who found advantages for a HNSSI group on an emotion recognition task, such as lower emotion intensity thresholds for negative emotions. One could argue that these advantages may be the result of the more dynamic and richer stimuli (moving, coloured) used, which more realistically mimicked in vivo social interactions compared to the static, grey-scale images distorted by fractal noise, which were used in the current study. These contradicting results may also suggest that differences exist at a higher order level of visual and cognitive processes (such as prior knowledge or memory) on this perceptual task.

To date, two studies have examined facial emotion recognition capabilities in individuals exhibiting NSSI behaviours. An analogous study that used a morphing emotion paradigm also found no group differences in the intensity of emotion required for correct participant categorization of happy, sad, angry, disgusted, fearful, or neutral facial expressions, nor did they see any group differences in the accuracy of emotion recognition [51]. However, this study also used sad and neutral mood induction to evoke specific mood states before presenting the stimuli, which was not the case for the current research or the Ziebell et al., [30] study. Seymour et al. [52], who also used static images for their experiment, found that adolescents engaging in NSSI made more errors on child fearful and adult sad face recognition compared to the typically developing controls.

According to Linehan's biosocial theory [53], the emotion dysregulation observed in individuals with Borderline Personality Disorder (BPD) is hypothesized to be a consequence of their greater emotional sensitivity. Although NSSI is frequently a symptom of BPD, NSSI and BPD can also occur separately and be conceptualized along a continuum with some phenomenological overlap [54–56]. The research findings on emotion recognition capabilities in adolescents with BPD are inconsistent and range from no differences [57–58], to decreased sensitivity to some facial emotions [59]. Studies examining adult participants with BPD are likewise inconsistent, with greater sensitivity to facial expressions reported [39] in addition to no differences [60]. Mitchell et al.'s [40] review of emotion recognition in BPD concluded no significant recognition impairments in BPD compared to controls for any negative emotions, despite the various methodological differences between studies. However, only a minority of individuals engaging in NSSI behavior meet the criteria for BPD [51, 61, 62]. These divergent results within and between BPD and NSSI studies, as well as differences in the studied populations, make it unclear as to what would be expected regarding emotion recognition capabilities from individuals who have a history of engaging in NSSI in the absence of BPD.

In the current study, differences in accuracy scores were observed, as assessed through the unbiased hit rate, which showed decreased accuracy for the HNSSI group in response to correctly categorizing fearful faces at 75% and 100% intensity. This trend was also observed at the lowest intensity of fear at 50%, which came out slightly above the conservatively adjusted p-value to accommodate for the multiple comparisons. These results appear in line with the observations made by Seymour et al., [52] who found that inpatient adolescents engaged in NSSI made more emotional face recognition errors for child fearful faces compared to typically developing controls. Among other reasons, theoretical models of NSSI postulate that engaging in NSSI may be used by individuals as a means for social communication to gain attention or influence other's behavior [63–64]. Moreover, other empirical studies have found that individuals who engage in NSSI display social communication skills deficits such as impaired social problem-solving, poor verbal skills, and alexithymia [11,19, 65]. Hence it is possible that individuals who have engaged in NSSI may indeed display deficits in the critical social communication skill of accurate emotional face recognition. Deficits in emotional face identification could also contribute to the mechanism by which feelings of social isolation develop and perpetuate the cycle of self-harm. If an individual who has engaged in NSSI misinterprets their surrounding social cues, such as fearful faces as our data suggests, then they may attribute incorrect emotional responses to those around them, leading to further social isolation and emotional activation. Research conducted by Marsh et al. [66] has indicated that the ability to recognize fearful facial expressions can predict prosocial behavior. Hence, a misinterpretation of fearful faces is likely to contribute to the emotional disturbances, inadequate social behavior, and less adaptive social problem-solving skills often observed in adolescents who engage in NSSI behavior [67, 65]. Of course, further empirical investigation is needed to help develop and validate this theory. A discrepancy in emotion identification abilities of HNSSI individuals

may also exist between their perceptions of dynamic images versus the static images presented in the current study. Moreover, the current study was conducted with individuals who reported a history of engaging in NSSI and were not necessarily actively engaged in self-injury at the time of study. Examining this population, although unique and previously unstudied, may also provide weaker results compared to the results of the aforementioned studies that investigated adolescents actively engaged in self-injury.

While an error pattern analysis across the six emotions provided some occasional significant results, few consistent patterns were clearly observed. Interesting observations were noted in errors made between sad and surprised expressions. In particular, HNSSI participants appeared significantly less likely to confuse sad facial emotions with surprise compared to control, particularly when errors were collapsed across the three emotion intensities. However, the clinical relevance of this finding and how it relates to HNSSI emotion perception remains somewhat unclear. A more easily interpreted finding is the HNSSI group's tendency to confuse angry expressions at low intensities with the emotion of happy, and happy expressions at a high intensity with the emotion of anger compared to controls. After collapsing across emotion intensity, the HNSSI group was significantly more likely to mistake angry faces for happy expressions. Confusion between positive and negative emotions, particularly happy and angry expressions, could easily lead to inappropriate responses in social situations. When social exchanges require the interpretation of subtle emotional cues, individuals who mistakenly perceive the valence of emotions may be particularly taxed when required to navigate emotionally charged situations. Work by Demers et al., [21], shows results consistent with the idea that adolescents engaged in NSSI show reduced vigilance to positive social information. This result, in combination with our study results, lends further support to the notion that individuals who have engaged in NSSI experience difficulty evaluating interpersonal situations [20].

In summary, this study found no significant group differences between HNSSI and controls participants regarding signal-proportion thresholds or efficiency scores. Results did, however, find decreased accuracy of HNSSI participants for recognizing fearful expressions and an increased likelihood of mistaking angry faces for happy expressions compared to controls. These findings provide further support for the literature that proposes deficits in accurate emotion perception of static images by individuals who have engaged in NSSI behaviours.

## Supporting information

**S1 Fig. Group differences in errors made for fear at 3 emotion intensities.**
(TIF)

**S2 Fig. Group differences in errors made for anger at 3 emotion intensities.**
(TIF)

**S3 Fig. Group differences in errors made for disgust at 3 emotion intensities.**
(TIF)

**S4 Fig. Group differences in errors made for happy at 3 intensities.**
(TIF)

**S5 Fig. Group differences in errors made for sad at 3 intensities.**
(TIF)

**S6 Fig. Group differences in errors made for surprise at 3 intensities.**
(TIF)

**S7 Fig. Group differences in errors made for anger collapsed across intensities.**
(TIF)

**S8 Fig. Group differences in errors made for happy collapsed across intensities.**
(TIF)

**S9 Fig. Group differences in errors made for sad collapsed across intensities.**
(TIF)

**S1 File. Supporting data and error analysis summary.** This supporting file contains the results of the mixed factorial ANOVA for threshold (Table A), efficiency (Table B), unbiased hit rate (Table C), and hit rate accuracy (Table D). It also contains the hit rate accuracy for recognition of fearful facial expressions across intensities (Table E), the Shapiro-Wilk tests of normality (Table F), skewness and kurtosis of error responses to presented emotions (Table G), and a detailed error analysis summary.
(DOCX)

**S2 File. Raw data.** This supporting file contains the threshold, efficiency and error raw data of the experiment.
(XLSX)

## Acknowledgments

The authors would like to acknowledge and thank the participants of this study for their time devoted to data collection.

## Author Contributions

**Conceptualization:** Laura Ziebell, Charles Collin, Stéphane Rainville.

**Data curation:** Laura Ziebell, Monica Mazalu, Madyson Weippert.

**Formal analysis:** Laura Ziebell, Stéphane Rainville.

**Funding acquisition:** Laura Ziebell, Charles Collin.

**Investigation:** Laura Ziebell.

**Methodology:** Laura Ziebell, Charles Collin, Stéphane Rainville.

**Project administration:** Laura Ziebell.

**Resources:** Charles Collin.

**Software:** Charles Collin.

**Supervision:** Charles Collin.

**Writing – original draft:** Laura Ziebell.

**Writing – review & editing:** Laura Ziebell, Charles Collin, Stéphane Rainville, Monica Mazalu, Madyson Weippert.

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
