## [Decision Letter · Decision Letter 0]

18 Oct 2019

PONE-D-19-21938

Using an Ideal Observer analysis to investigate the visual perceptual efficiency of individuals with a history of NSSI when identifying emotional expressions

PLOS ONE

Dear Ms. Ziebell,

Thank you for submitting your manuscript to PLOS ONE. After careful consideration, we feel that it has merit but does not fully meet PLOS ONE’s publication criteria as it currently stands. Therefore, we invite you to submit a revised version of the manuscript that addresses the points raised during the review process.

As noted below by the reviewer, the manuscript was very well written and addressed an important topic for the field. There are a few minor analyses that could be conducted to clarify the findings presented in the paper. In addition to those presented by the reviewer below, I would like to see whether there were any statistically significant group differences between the HNSSI and Control groups on demographic characteristics. Additional reviewer suggestions for further revision can be found at the end of this message.

We would appreciate receiving your revised manuscript by Dec 02 2019 11:59PM. To enhance the reproducibility of your results, we recommend that if applicable you deposit your laboratory protocols in protocols.io, where a protocol can be assigned its own identifier (DOI) such that it can be cited independently in the future. For instructions see: http://journals.plos.org/plosone/s/submission-guidelines#loc-laboratory-protocols

We look forward to receiving your revised manuscript.

Kind regards,

Sarah A. Arias

Academic Editor

PLOS ONE

Journal Requirements:

3. Please state in your methods section whether you obtained consent from parents or guardians of the minors included in the study (participants aged under 18) or whether the research ethics committee or IRB approved the lack of parent or guardian consent.

4. Please note that according to our submission guidelines (http://journals.plos.org/plosone/s/submission-guidelines), outmoded terms and potentially stigmatizing labels should be changed to more current, acceptable terminology. For example: “Caucasian” should be changed to “white” or “of [Western] European descent” (as appropriate).

Please consider defining the acronym "NSSI" in the title.

5. We note that Figure 1 in your submission contain copyrighted images. All PLOS content is published under the Creative Commons Attribution License (CC BY 4.0), which means that the manuscript, images, and Supporting Information files will be freely available online, and any third party is permitted to access, download, copy, distribute, and use these materials in any way, even commercially, with proper attribution. For more information, see our copyright guidelines: http://journals.plos.org/plosone/s/licenses-and-copyright.

6. Thank you for stating the following in the Financial Disclosure section:

This work was supported by funding from the Canadian Institutes of Health Research (http://www.cihr-irsc.gc.ca/e/193.html) to LZ under Grant 201310GAD; and the Natural Sciences and Engineering Research Council of Canada (http://www.nserc-crsng.gc.ca/index_eng.asp) to CC under Grant 2015-05067. The funders had no role in study design, data collection and analysis, decision to publish, or preparation of the manuscript.

We note that one or more of the authors are employed by a commercial company: VizirLabs Consulting,

Reviewers' comments:

Reviewer's Responses to Questions

**Comments to the Author**

1. Is the manuscript technically sound, and do the data support the conclusions?

Reviewer #1: Yes

2. Has the statistical analysis been performed appropriately and rigorously? 

Reviewer #1: Yes

3. Have the authors made all data underlying the findings in their manuscript fully available?

Reviewer #1: Yes

4. Is the manuscript presented in an intelligible fashion and written in standard English?

Reviewer #1: Yes

5. Review Comments to the Author

Reviewer #1: Thank you for the opportunity to review this well-written and fascinating manuscript, in which the authors make a substantial contribution to the literature on emotion processing in NSSI. In their investigation, the authors found no differences between college students with and without NSSI history in the threshold nor efficiency of emotion recognition, although they observed differences in accuracy of detecting negative emotional expressions. Specifically, the group with NSSI history more frequently mis-classified angry and sad faces as depicting joy and surprise, respectively.

The paper is very clear throughout, and I appreciate the simplicity with which the authors were able to convey important information about Ideal Observer analyses. Their analytic plan was thorough and I applaud the rigor of their study design. Overall this is a strong work that I recommend for publication in PLOS ONE; I have a few minor suggestions for exploratory analyses that the authors may consider:

I agree with the authors’ decision to exclude participants based on BPD diagnoses. I am curious to know whether the authors examined linear relations with BPD traits or NSSI history (and the dependent variables of interest)? For example, do individuals with more "severe" NSSI histories, based on frequency or recency, demonstrate emotion recognition threshold or efficiency differences? Our recent work found hypothesized effects in an emotional inhibition task (specifically, a “directed forgetting” paradigm) only among a subgroup of participants with NSSI history who also reported elevated BPD traits (see Best, Allen, & Hooley, 2019). I therefore wonder if a comparable BPD effect may be operating in these data.

I also wish to know whether the authors tried any additional transformations (e.g., log) to improve data normality in the emotion recognition accuracy analyses (besides the arcsine transform)?

6. PLOS authors have the option to publish the peer review history of their article (what does this mean?). If published, this will include your full peer review and any attached files.

Reviewer #1: Yes: Kenneth J.D. Allen, Ph.D.

---

## [Author Response · Author response to Decision Letter 0]

2 Dec 2019

Laura Ziebell

University of Ottawa, Perception and Cognition Lab (PCL)

Vanier 3089, 136 J-J Lussier Pvt.

Ottawa, ON, K1N 6N5

lzieb079@uottawa.ca

Sarah A. Arias

Academic Editor

PLOS ONE

Dec 1, 2019

Dear Ms. Arias,

Thank you kindly to the reviewer and yourself for your considered and valuable feedback on this manuscript. Please see below for a response to comments and revisions.

Journal Requirements:

Response: The links provided for the style guides were consulted in detail, and every effort has been made to comply with the style templates for the title page and body of the manuscript, including file naming. 

Response: Captions for the Supporting Information files have been added at the end of the manuscript, and in-text citations were updated. 

3. Please state in your methods section whether you obtained consent from parents or guardians of the minors included in the study (participants aged under 18) or whether the research ethics committee or IRB approved the lack of parent or guardian consent.

Response: A statement was included in the methods section stipulating that the research ethics committee approved the participation of minors in this study without parental or guardian consent. 

4. Please note that according to our submission guidelines (http://journals.plos.org/plosone/s/ submission-guidelines), outmoded terms and potentially stigmatizing labels should be changed to more current, acceptable terminology. For example: “Caucasian” should be changed to “white” or “of [Western] European descent” (as appropriate). Please consider defining the acronym "NSSI" in the title.

Response: The terminology has been updated and “Caucasian” has been changed to “white”. Also, the acronym "NSSI" has been defined as non-suicidal self-injury in the title.

5. We note that Figure 1 in your submission contain copyrighted images. All PLOS content is published under the Creative Commons Attribution License (CC BY 4.0), which means that the manuscript, images, and Supporting Information files will be freely available online, and any third party is permitted to access, download, copy, distribute, and use these materials in any way, even commercially, with proper attribution.

Response: According to the website stipulating the use and publication of the KDEF (see: http://kdef.se/home/using%20and%20publishing%20kdef%20and%20akdef.html), “Researchers may always include sample images from KDEF in his/her manuscript when said manuscript is a doctoral thesis OR is a manuscript submitted to a scientific journal. A publisher may regard this mail as a written consent for such publication or contact me (daniel.lundqvist@ki.se) directly if needed. For the KDEF stimuli, such a journal is typically PLOS ONE, EMOTION, COGNITION & EMOTION, NEUROPSYCHOLOGIA, SOCIAL COGNITIVE & AFFECTIVE NEUROSCIENCE, BIOLOGICAL PSYCHOLOGY, NEUROIMAGE, FRONTIERS IN PSYCHOLOGY, JOURNAL OF NEUROSCIENCE or PSYCHONEUROENDOCRINOLOGY or similar.”

Additionally, Daniel Lundqvist from the Karolinska Institutet, who created the KDEF materials together with colleagues in 1998 and maintains the KDEF website, was contacted directly for permission to reprint the image before submitting to PLOS ONE. See written permission provided by Daniel Lundqvist attached as an "Other" file to this submission. 

6. We note that one or more of the authors are employed by a commercial company: VizirLabs Consulting. Please provide an amended Funding Statement declaring this commercial affiliation, as well as a statement regarding the Role of Funders in your study. If the funding organization did not play a role in the study design, data collection and analysis, decision to publish, or preparation of the manuscript and only provided financial support in the form of authors' salaries and/or research materials, please review your statements relating to the author contributions, and ensure you have specifically and accurately indicated the role(s) that these authors had in your study. You can update author roles in the Author Contributions section of the online submission form.

Response: The author, Stéphane Rainville, is the founder and sole employee of VizirLabs Consulting. No funding was provided to this research study by VizirLabs Consulting and the author had no vested interest in the outcome of this research. Dr. Rainville collaborated on this project by providing programming for the experiment and data aggregation, as directed by the primary author. An amended Funding Statement declaration will stipulate that funding not was provided by this commercial affiliation (see below). Dr. Rainville contributed to the publication by provided feedback on the description of the ideal observer performance in the procedure section. 

The Financial Disclosure section now reads as:

This work was supported by funding from the Canadian Institutes of Health Research (http://www.cihr-irsc.gc.ca/e/193.html) to LZ under Grant 201310GAD; and the Natural Sciences and Engineering Research Council of Canada (http://www.nserc-crsng.gc.ca/index_eng.asp) to CC under Grant 2015-05067. The funders had no role in study design, data collection and analysis, decision to publish, or preparation of the manuscript. VizirLabs Consulting did not provide any financial contributions to this research. 

Competing Interests Statement now reads:

Affiliation with VizirLabs Consulting does not alter our adherence to PLOS ONE policies on sharing data and materials. Stephane Rainville was paid to assist in programming the experiment and for creating scripts that aided in data aggregation. He did so based on instructions provided from the primary author.

Reviewer Comments:

7. Reviewer #1: Thank you for the opportunity to review this well-written and fascinating manuscript, in which the authors make a substantial contribution to the literature on emotion processing in NSSI. In their investigation, the authors found no differences between college students with and without NSSI history in the threshold nor efficiency of emotion recognition, although they observed differences in accuracy of detecting negative emotional expressions. Specifically, the group with NSSI history more frequently mis-classified angry and sad faces as depicting joy and surprise, respectively.

The paper is very clear throughout, and I appreciate the simplicity with which the authors were able to convey important information about Ideal Observer analyses. Their analytic plan was thorough, and I applaud the rigor of their study design. Overall this is a strong work that I recommend for publication in PLOS ONE; I have a few minor suggestions for exploratory analyses that the authors may consider:

I agree with the authors’ decision to exclude participants based on BPD diagnoses. I am curious to know whether the authors examined linear relations with BPD traits or NSSI history (and the dependent variables of interest)? For example, do individuals with more "severe" NSSI histories, based on frequency or recency, demonstrate emotion recognition threshold or efficiency differences? Our recent work found hypothesized effects in an emotional inhibition task (specifically, a “directed forgetting” paradigm) only among a subgroup of participants with NSSI history who also reported elevated BPD traits (see Best, Allen, & Hooley, 2019). I therefore wonder if a comparable BPD effect may be operating in these data.

I also wish to know whether the authors tried any additional transformations (e.g., log) to improve data normality in the emotion recognition accuracy analyses (besides the arcsine transform)?

Response: We thank the reviewer for their complimentary commentaries, as well as their valuable and thoughtful suggestions. 

The reviewer’s suggestion to examine a linear relationship with BPD traits or NSSI history with the dependent variables of interest is a thought-provoking suggestion. However, the number of participants recruited for this study would not provide sufficient statistical power to reliably conduct and draw conclusions from a linear relations analysis. Such an analysis would require a great deal of further data collection with a much larger sample size. Additionally, information regarding specific BPD traits were not collected as part of this research design. These suggestions, however, would be interesting to explore in future research. 

The authors did attempt to improve data normality with several other transformations, such as log10 and square root transformations, in addition to the arcsine transformation suggested by Wanger (1993) for unbiased hit rate analysis. However, these transformations did not sufficiently improve data normality. Consequently, the decision was made to continue analysis with the untransformed data. This information has been added to the manuscript. 

Editor Comments:

8. In addition to those presented by the reviewer below, I would like to see whether there were any statistically significant group differences between the HNSSI and Control groups on demographic characteristics. 

Response: Analyses were completed to determine statistically significant group differences between the HNSSI and Control groups on demographic characteristics and included in the Materials and Methods section. 

We hope these modifications and responses have adequately addressed the matters highlighted by the reviewer and editor. It is our hope that the manuscript is now fit for publication in PLOS ONE. 

Thank you again kindly for your consideration!

Sincerely,

Laura Ziebell, M.Sc. Behavioural Neuroscience

Ph.D. Candidate Clinical Psychology

University of Ottawa / Université d'Ottawa

---

## [Editor Report · Decision Letter 1]

12 Dec 2019

Using an Ideal Observer analysis to investigate the visual perceptual efficiency of individuals with a history of NSSI when identifying emotional expressions

PONE-D-19-21938R1

Dear Dr. Ziebell,

We are pleased to inform you that your manuscript has been judged scientifically suitable for publication and will be formally accepted for publication once it complies with all outstanding technical requirements.

With kind regards,

Sarah A. Arias, PhD

Academic Editor

PLOS ONE

---

## [Editor Report · Acceptance letter]

13 Jan 2020

PONE-D-19-21938R1 

Using an Ideal Observer analysis to investigate the visual perceptual efficiency of individuals with a history of NSSI when identifying emotional expressions 

Dear Dr. Ziebell:

I am pleased to inform you that your manuscript has been deemed suitable for publication in PLOS ONE. Congratulations! Your manuscript is now with our production department. 

With kind regards,

on behalf of

Dr. Sarah A. Arias 

Academic Editor

PLOS ONE